

# Gait and dynamic pedobarographic analyses in hallux rigidus patients treated with Keller's arthroplasty, arthrodesis or cheilectomy 22 years after surgery

Robin de Bot[1,2,3], Jasper Stevens[1,2,3], Thijs Smeets[1,4], Adhiambo Witlox[2], Wieske Beertema[1], Roel Hendrickx[1], Kenneth Meijer[3] and Martijn Schotanus[1,2,5]

[1] Department of Orthopaedic Surgery and Traumatology, Zuyderland Medical Center, Geleen, Limburg, The Netherlands
[2] Department of Orthopaedic Surgery, Maastricht University Medical Center, Maastricht, Limburg, The Netherlands
[3] Department of Nutrition and Movement Sciences, NUTRIM School of Nutrition and Translational Research in Metabolism, Maastricht University, Maastricht, Limburg, The Netherlands
[4] Smeets Loopcomfort, Sittard, Limburg, The Netherlands
[5] School of Care and Public Health Research Institute, Faculty of Health, Medicine & Life Sciences, Maastricht University Medical Center, Maastricht, Limburg, The Netherlands

Corresponding author
Robin de Bot, rtaldebot@gmail.com

## ABSTRACT

**Background**. Keller's arthroplasty, arthrodesis and cheilectomy are well-known surgical interventions for hallux rigidus. This study aimed to evaluate the effects of these surgical interventions on gait, plantar pressure distribution and clinical outcome in patients treated for hallux rigidus 22 years after surgery.

**Methods**. Spatio-temporal gait parameters and plantar pressure distribution, determined as pressure time integrals (PTIs) and peak pressures (PPs), were analyzed using a 7-foot tone analysis model. Patient-reported outcome was assessed using the Manchester-Oxford Foot Questionnaire (MOXFQ). Of the 73 patients (89 feet) from the original study, 27 patients (33 feet) and 13 healthy controls (26 feet) were available for evaluation 22 years after hallux rigidus surgery.

**Results**. Spatio-temporal gait parameters were comparable between all groups and were in line with healthy controls ($P > 0.05$). No differences ($P > 0.05$) in PTIs and PPs were found in the seven plantar zones between groups and as compared to healthy controls. MOXFQ scores in all domains (walking/standing, range 21.4–24.1; pain, range 16.5–22.2 and social interaction, range 23.8–35.4) were not clinically and statistically different ($P > 0.05$) between the three different surgical interventions.

**Conclusion**. These results suggest no long-term functional and biomechanical differences after these surgical interventions for hallux rigidus correction. The interventions seem to be appropriate treatment options for a selective group of patients with symptomatic hallux rigidus.

## INTRODUCTION

Hallux rigidus (HR) defined as osteoarthritis of the first metatarsophalangeal (MTP1) is the most commonly affected joint in patients with osteoarthritis of the foot (*Roddy & Menz, 2018*). Most reported symptoms are pain, swelling and a restriction of range of motion (RoM) of the MTP1 joint leading to difficulties while walking (*Coughlin & Shurnas, 2003a*; *Galois et al., 2019*). These symptoms progress over time resulting in a significantly decreased experienced health-related quality of life (*Gilheany, Landorf & Robinson, 2008*).

Surgical interventions are indicated when conservative treatments failed (*Galois et al., 2019*). Keller's arthroplasty, arthrodesis and cheilectomy are surgical interventions for HR (*Galois et al., 2019*; *Ho & Baumhauer, 2017*). In Keller's arthroplasty, the base of the proximal phalanx is resected leading to MTP1 joint decompression and increased dorsiflexion. In this procedure, joint stability is sacrificed, which results in an unstable and non-supporting hallux (*Galois et al., 2019*; *Yee & Lau, 2008*). An arthrodesis, in which the first metatarsal bone and proximal phalanx are fused, leads to a stiff and motionless MTP1 joint (*Alentorn-Geli et al., 2013*; *Coughlin & Shurnas, 2003a*; *Galois et al., 2019*; *Stevens et al., 2017a*; *Yee & Lau, 2008*). In cheilectomy, the dorsal osteophyte at the first metatarsal head is removed, leading to a reduction in pain and improvement in RoM after surgery (*Galois et al., 2019*; *Yee & Lau, 2008*). Cheilectomy is predominantly recommended for patients with mild to moderate HR, while Keller's arthroplasty and arthrodesis is performed in patients with a more progressed stage of HR (stage 2 and 3) (*Galois et al., 2019*; *Ho & Baumhauer, 2017*; *Yee & Lau, 2008*). Improvements in clinical outcome and patient satisfaction are reported after all three surgical techniques (*Beertema et al., 2006*; *Coughlin & Shurnas, 2003b*; *Schneider et al., 2011*; *Stevens et al., 2017a*).

Several studies have been performed to evaluate clinical outcomes after these surgical interventions. Only a few studies have evaluated gait and plantar pressures distribution after Keller's arthroplasty (*Schneider et al., 2011*), arthrodesis (*Chraim et al., 2016*; *DeFrino et al., 2002*; *Gibson & Thomson, 2005*; *Stevens et al., 2017b*) and cheilectomy (*Lau & Daniels, 2001*; *Nawoczenski, Ketz & Baumhauer, 2008*) for HR. None of these studies directly compared these techniques. Previous mentioned studies observed reduced peak pressures under the hallux whereas increased peak pressures were observed beneath the first to the fourth metatarsal head after surgery when compared to unaffected feet (*Lau & Daniels, 2001*; *Schneider et al., 2011*; *Stevens et al., 2017b*). However, a high methodological heterogeneity is present between these studies with respect to: (i) measurements systems, (ii) used pressure distribution models and subdivision in foot areas, (iii) used control groups (iv) variety in outcomes and (v) differences in follow-up period (*Chraim et al., 2016*; *DeFrino et al., 2002*; *Gibson & Thomson, 2005*; *Lau & Daniels, 2001*; *Nawoczenski, Ketz & Baumhauer, 2008*; *Schneider et al., 2011*; *Stevens et al., 2017b*). Currently, no long-term evaluation studies are available that evaluated long-term effects of these interventions for HR on biomechanical outcomes.

The aim of this comparative long-term evaluation study was to evaluate the effects of Keller's arthroplasty, arthrodesis and cheilectomy on gait, plantar pressure distribution and clinical outcome of patients treated for HR 22 years after surgery. Results were compared to

each other and compared to healthy controls without foot complains. Correlation analysis was performed between gait and clinical outcomes to observe if certain associations could be detected. It was hypothesized that after long-term follow-up, a high satisfaction and normalized plantar pressures beneath the 1th and 2nd metatarsal head will be observed in subjects with an arthrodesis compared to healthy controls. Since, the MTP1-joint is sacrificed and metatarsal bones are fused, which creates stability and convert soft tissue and intrinsic foot musculature into stabilizing forces resulting in reestablishment of weight bearing distribution of the foot. Keller's arthroplasty will follow this distribution, since motion and some rolling of is still possible within the unstable and non-supporting hallux. Finally, the lowest pressures are expected for the cheilectomy group since, they will avoid the medial side of the forefoot as a result of the expected disease progression, and accompanying pain.

## METHODS

### Study participants

Eligible study participants were derived from a cohort previously evaluated after a follow-up period of 7-years after surgery (*Beertema et al., 2006*). Patients underwent Keller's arthroplasty, arthrodesis or cheilectomy operations between 1990 and 2000 for a symptomatic HR. All patients had pain and loss of motion of the MTP1 joint (*Beertema et al., 2006*). The operations were performed by four experienced surgeons, who used a consistent operative technique and standardized postoperative regimen for each procedure. Participants who were able to walk barefoot and participated in the previous study were eligible for inclusion (*Beertema et al., 2006*). Furthermore, results of the interventions groups were compared to healthy controls with a comparable median age, sex and body mass index (BMI). The same inclusion criteria were eligible for the healthy controls. Healthy controls were free of any clinical signs or symptoms of hallux rigidus or other pathological conditions of the lower extremities.

### Study design

This study was performed at our department of Orthopaedic Surgery and Traumatology. Long-term clinical outcome of these patients were recently evaluated (*Stevens et al., 2020*). A clinical examination (*i.e.,* anthropometrics), gait analysis (*i.e.,* spatio-temporal gait characteristics and pedobarographic assessment) and patient-reported outcome measures (PROMs) was performed at follow-up. Results of the present study were not compared to the previous study since the outcomes of the present study were not evaluated in the initial study (*Beertema et al., 2006*). The study was performed according to the Declaration of Helsinki (2013) and the Medical Ethical Committee Zuyderland (number 17-T-09) gave approval for this study. All patients provided written informed consent.

### Gait and pedobarographic analysis

The Zebris FDM-TLR instrumented treadmill (Zebris Medical GmbH, Isny, Germany) was used for gait and pedobarographic analysis. This treadmill is equipped with a 94.8 cm × 40.6 cm electronic mat sensor (Zebris Medical GmbH, Isny, Germany) embedded

beneath the belt. It contains 5,376 miniature capacitive pressure sensors, registering the exerted force at a rate of 100 Hz ranging from 1 to 120 N/cm$^2$. The speed of the treadmill can be adjusted from 0.8 up to 14 km/h with intervals of 0.1 km/h. Patients started with walking at the treadmill for four minutes to become familiarized with it. After this period of acclimatization to the treadmill, subjects were asked to walk at a self-selected comfortable speed, which is essential to obtain comparable data to overground walking (*Watt et al., 2010*). To determine comfortable walking speed, participants started walking at a fixed speed of 0.5 km/h. Subsequently, belt speed was increased in a stepwise manner with steps varying form 0.1 to 0.3 km/h until the comfortable walking speed was reached. Thereafter, two measurements were performed, resulting in 30 to 40 steps per measurement. One of these two measurements were randomly chosen and used for further analysis.

The integrated 7-foot tone analysis model WinFDM-T software version 2.0.39 (Zebris medical GmbH, Isny, Germany) was used to assess gait and pedobarographic data. This software divides the foot in seven zones; heel lateral, heel medial, midfoot, forefoot lateral, forefoot inner, forefoot medial and toes (Fig. 1). In this model, it was suggested that the forefoot medial represents the 1th and 2th metatarsal head, forefoot inner the 3th and 4th metatarsal head and forefoot lateral the 5th metatarsal head. All toes (*i.e.,* 1 to 5) were included in the toe zone. Spatio-temporal parameters of interest were gait velocity (km/h), step length (cm), step width (cm), step time (s), stance phase (%; subdivided in load response, mid stance and pre-swing), swing phase (%) and double stance phase (%). For plantar pressure analysis, the pressure time integral (PTI; Ns/cm$^2$) and peak pressures (PP; N/cm$^2$) of the pressure curves were determined using a software tool developed by using MATLAB (MathWorks, version 9.7, Natick, MA, USA) (*Keijsers, Stolwijk & Pataky, 2010*; *Melai et al., 2011*). Both outcomes are mainly used in pedobarographic studies (*Keijsers, Stolwijk & Pataky, 2010*; *Melai et al., 2011*; *Orlin & McPoil, 2000*). The PTI described the cumulative effect of pressure over time in a certain area of the foot and therefore provided a value for the total load exposure of a planter area during stance (*Keijsers, Stolwijk & Pataky, 2010*; *Melai et al., 2011*). The peak pressure was the maximum peak pressure measured in one zone during the stance phase (*Keijsers, Stolwijk & Pataky, 2010*; *Melai et al., 2011*; *Orlin & McPoil, 2000*).

The foot specific PROMs, the Manchester-Oxford Foot Questionnaire (MOXFQ), which is often used to assess clinical outcome in hallux surgery, was used (*Dawson et al., 2007*; *Venkatesan, Schotanus & Hendrickx, 2016*). The MOXFQ is used since it gave insights in the experience and satisfaction of patients and evaluates three domains; walking/standing problems (seven items), foot pain (five items) and issues related to social interaction (four items). A score of 0 represents the best outcome and 100 as the poorest outcome (*Venkatesan, Schotanus & Hendrickx, 2016*).

## Statistical analysis

Statistical analysis was conducted with GraphPad Prism 8.3 (Graphpad Software, Inc., version 9.1.1., San Diego, CA, USA) and SPSS (IBM Statistics, version 25, Armonk, NY, USA). Descriptive statistics were calculated for patient demographics, spatio-temporal parameters, plantar pressure measurement, and PROMs. Normality of distributions was
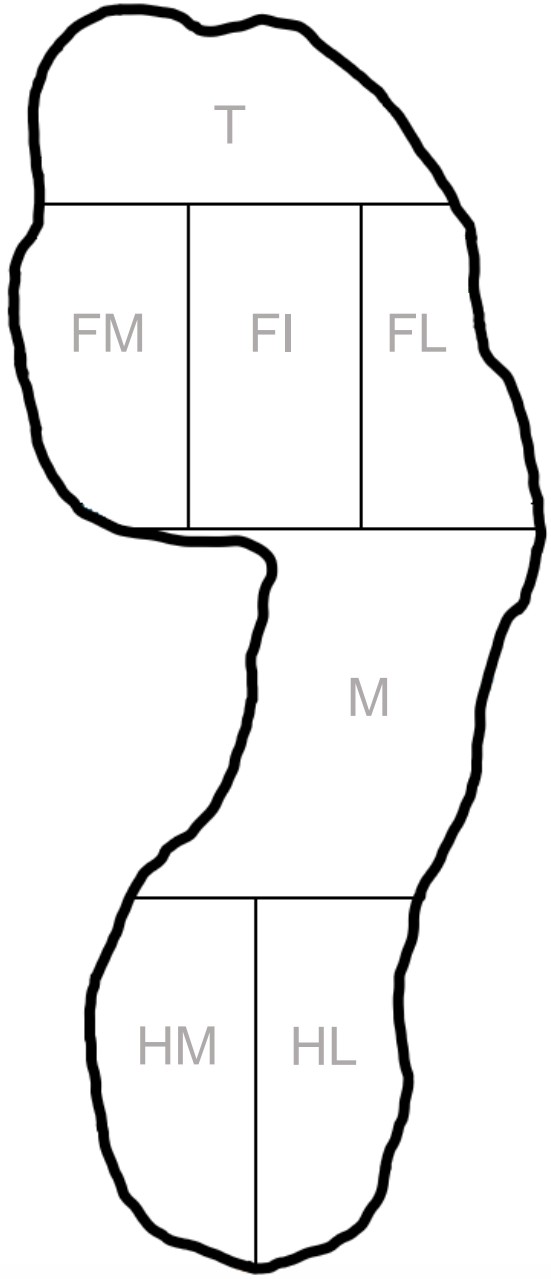

**Figure 1 Seven plantar area's according to the Zebris 7-foot tone model.** The foot was divided into toes (T), forefoot medial (FM), forefoot inner (FI), forefoot lateral (FL), midfoot (M), heel medial (HM) and heel lateral (HL).

tested using the D'Agostino-Pearson normality test and revealed that spatio-temporal parameters and plantar pressure data need to be analyzed using non-parametric statistics. The Kruskal-Wallis test was conducted to test for significant differences between the three intervention groups and control group. Dunn's multiple comparison test was performed

for pairwise comparisons (*Dunn, 1964*). Effect sizes were calculated using the Hodges-Lehmann estimator of location shift, calculated as the median of differences. According to the D'Agostino-Pearson normality test, there was no significant departure of normality of the MOXFQ data. Therefore, an analysis of variance (ANOVA) was performed to test for differences between groups. Finally, Spearman rho correlation analysis was used to detect associations between plantar pressure in the forefoot and MOXFQ. A *P*-value ≤0.05 was considered as statistically significant.

# RESULTS

The initial study (2006) included 73 patients (89 ft) and after a median follow-up period of 22 years (range 19.0 up to 26.3 years), 27 patients (33 ft) were available for assessment. Results were compared to 13 (26 ft) healthy controls (Table 1). Of the 73 patients (89 ft), 17 patients (23%; 20 ft) died, six patients (8%; 6 ft) were lost to follow-up and 23 patients (32%; 30 ft) were not able or willing to participate in the present study resulting in 27 patients (37%; 33 ft) available for gait evaluation (*Beertema et al., 2006*). Of these patients, twelve subjects were treated with a Keller's arthroplasty (14 ft), eight with an arthrodesis (10 ft) and seven with a cheilectomy (9 ft). In each group, two patients underwent bilateral surgery. Based on preoperative radiographs, cheilectomy was performed in grade 1 and 2 HR and Keller's arthroplasty and arthrodesis predominantly in grade 2 and 3 HR, according to the Regnauld's classification (Table 1) (*Regnauld, 1986*). No statistically significant differences in demographical parameters were observed between the groups (Table 1).

## Gait analysis
No differences in gait velocity and spatio-temporal parameters were detected between intervention groups and healthy controls ($P > 0.05$) (Table 2).

## PTIs and PPs in foot zones
Analyses of the PTIs showed no statistically significant differences between the Keller's arthroplasty (KA), arthrodesis (A), cheilectomy (C), and control group (CG) in each of the seven analyzed foot zones (Fig. 2). Effect sizes calculated as Hodges-Lehmann median of differences ranged in the forefoot medial from 0.46 (CG *vs.* C, $P > 0.99$) to 2.75 Ns/cm$^2$ (A *vs.* C, $P = 0.54$), in forefoot inner from 0.21 (CG *vs.* C, $P > 0.99$) to 1.27 Ns/cm$^2$ (A *vs.* C, $P > 0.99$), forefoot lateral from 0.08 (CG *vs.* KA, $P > 0.99$) to 1.50 Ns/cm$^2$ (C *vs.* KA, $P = 0.77$), and toes from 0.48 (CG *vs.* C, $P > 0.99$) to 1.19 Ns/cm$^2$ (KA *vs.* C, $P > 0.99$).

Also, for PPs, no statistically significant differences were detected between the surgical and control groups in each of the foot zones of interest (Fig. 3). The largest differences were observed in the forefoot medial and forefoot inner. Effect sizes calculated as Hodges-Lehmann median of differences ranged in the forefoot medial from 1.14 (CG *vs.* C, $P > 0.99$) to 10.33 N/cm$^2$ (A *vs.* C, $P = 0.22$) and forefoot inner from 0.65 (CG *vs.* C, $P > 0.99$) to 12.63 N/cm$^2$ (A *vs.* C, $P = 0.46$).

**Table 1 Patient demographics[a].**

|  | Keller's arthroplasty | Arthrodesis | Cheilectomy | Healthy controls |
|---|---|---|---|---|
| Number of participants (number of feet) | 12 (14) | 8 (10) | 7 (9) | 13 (26) |
| Male/Female | 9/3 | 3/5 | 4/3 | 6/7 |
| Left feet/Right feet | 8/6 | 4/6 | 5/4 | 13/13 |
| Age at surgery (years) | 54.0 (48.5–58.3) | 48.5 (43.5–54.0) | 53.0 (50.0–56.0) | – |
| Age at follow-up (years) | 75 (71–78) | 71 (65–74) | 70 (68–84) | 68 (62-73) |
| Follow-up (years) | 22 (20-22.8) | 22 (19.5–26.3) | 22 (19–25) | – |
| Height (m) | 1.72 (1.64–1.73) | 1.67 (1.51–1.74) | 1.72 (1.64–1.79) | 1.75 (1.72-1.79) |
| Body mass (kg) | 83.0 (67.0–86.0) | 81.5 (65.0–93.0) | 80.0 (74.0–87.0) | 76.0 (70.0-89.5) |
| BMI (kg/m$^2$) | 27.8 (24.4–30.7) | 28.7 (26.8–37.8) | 28.4 (26.5–29.1) | 25.6 (23.2-27.5) |
| HR grade before surgery | Gr I: 2<br>Gr II: 8<br>Gr III: 3<br>N/A: 1 | Gr I:1<br>Gr II: 5<br>Gr III: 3<br>N/A: 1 | Gr I: 5<br>Gr II: 2<br>Gr III: 0<br>N/A: 2 | – |

**Notes.**
[a]Median and interquartile range are presented in parentheses.
*No statistically significant differences were detected between the groups $P > 0.05$.
  BMI, Body mass index; N/A, no preoperative radiographic results were available.

**Table 2 Gait analysis results[a].**

|  | Keller's arthroplasty | Arthrodesis | Cheilectomy | Healthy controls |
|---|---|---|---|---|
| Gait velocity (km/h) | 2.5 (1.9–3.4) | 3.1 (1.1–4.5) | 2.8 (1.5–4.2) | 2.9 (2.6-3.9) |
| Step length (cm) | 41.6 (29.6–50.9) | 47.4 (37.7–58.2) | 47.9 (35.2–58.6) | 47.1 (43.5-51.4) |
| Step width (cm) | 10.2 (6.6–13.8) | 8.3 (6.0–14.5) | 7.7 (4.2–10.3) | 9.9 (6.4-14.2) |
| Step time (s) | 0.57 (0.51–0.76) | 0.55 (0.50–0.60) | 0.63 (0.55–0.76) | 0.55 (0.53-0.60) |
| Stance phase (%) | 65.7 (64.8–70.5) | 63.4 (62.7–65.7) | 65.5 (62.2–68.4) | 65.2 (62.8-66.5) |
| Load response (%) | 16.0 (14.5–21.4) | 13.4 (12.5–17.2) | 16.4 (12.1–17.4) | 15.0 (13.1-16.7) |
| Mid stance (%) | 33.8 (29.7–36.0) | 35.1 (33.2–37.3) | 34.2 (31.3–38.1) | 34.8 (33.4-37.3) |
| Pre-swing (%) | 16.0 (14.5–20.3) | 13.9 (12.5–18.1) | 16.8 (12.1–18.6) | 15.0 (13.0-16.8) |
| Swing phase (%) | 34.3 (29.5–35.2) | 36.6 (34.3–37.3) | 34.5 (31.6–37.8) | 34.8 (33.5-37.2) |
| Double stance phase (%) | 32.0 (30.0–36.3) | 29.2 (25.2–47.4) | 33.4 (24.1–38.2) | 30.4 (25.5-33.8) |

**Notes.**
[a]Median and 95% confidence interval presented in parentheses.
*No statistically significant differences were detected between the groups $P > 0.05$.

## PROMs

The lowest MOXFQ scores in all domains were reported in the arthrodesis group (walking/standing 21.4; pain 16.5 and social interaction 23.8), without statistically significant differences between the surgical groups ($P > 0.05$) (Table 3).

## Correlation analysis

Spearman rho correlation analysis showed no significant associations in the arthrodesis and Keller's arthroplasty group between the PTI in the medial forefoot and MOXFQ pain domain (A $r = -0.20$, $P = 0.58$; KA $r = -0.26$, $P = 0.42$) and MOXFQ walking/standing domain (A $r = -0.29$, $P = 0.42$; KA $r = -0.358$, $P = 0.25$). Only negative associations were observed after cheilectomy (PTI medial forefoot and MOXFQ pain $r = -0.78$ $P = 0.01$;

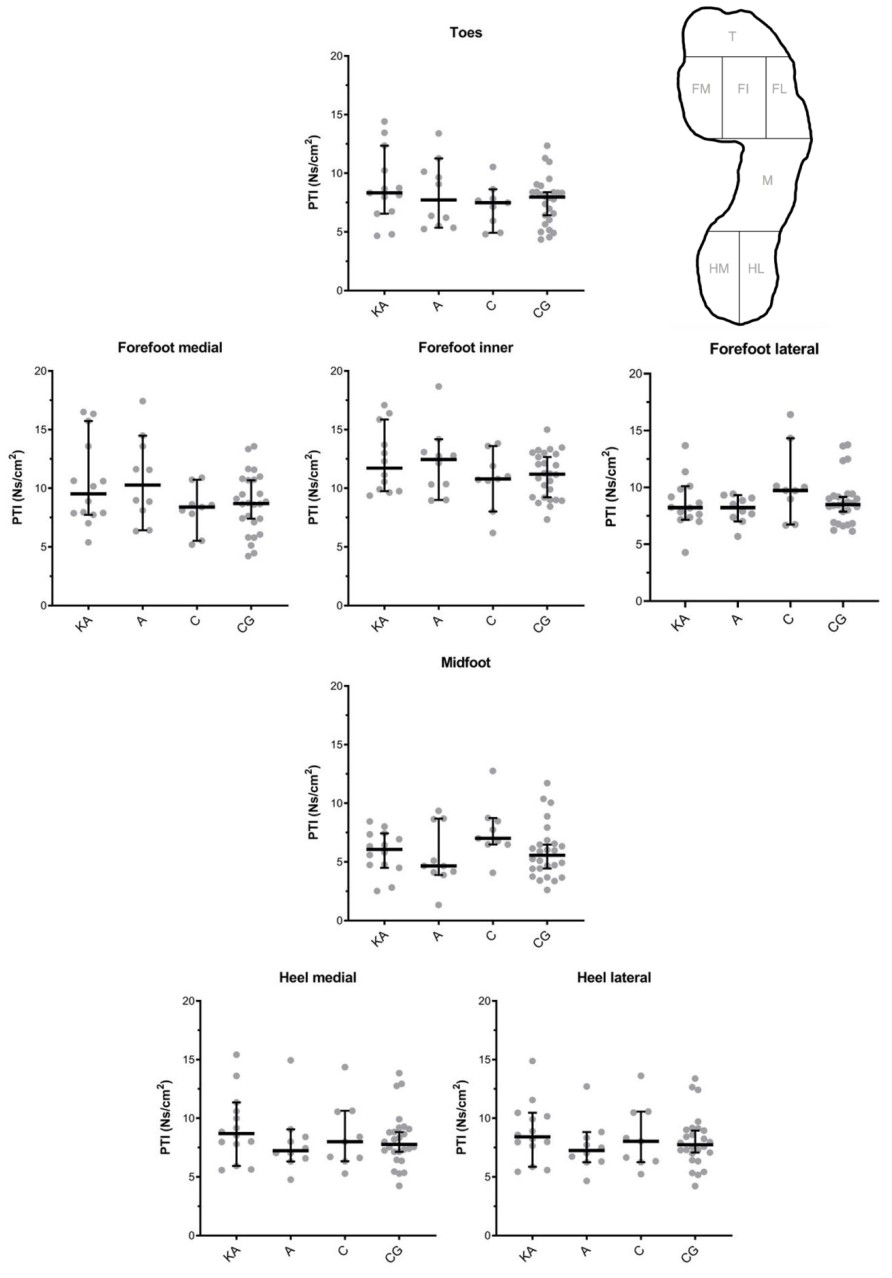

**Figure 2** Pressure time integrals (PTIs) of the Keller's arthroplasty (KA), arthrodesis (A), cheilectomy (C) and control group (CG) in seven foot zones as shown in Fig. 1. Results were presented as median and 95% confidence interval, with individual values. * All interventions were within each foot zone statistically compared to each other (*i.e.,* KA *vs.* A, KA *vs.* C, KA *vs.* CG, A *vs.* C, A *vs.* CG, C *vs.* CG) and no significant differences were detected ($P > 0.05$).

PTI medial forefoot and MOXFQ walking/standing $r = -0.69$, $P = 0.04$). No associations were observed after comparing PTI in the inner forefoot and PTI in the lateral forefoot to both MOXFQ domains in all three intervention groups.

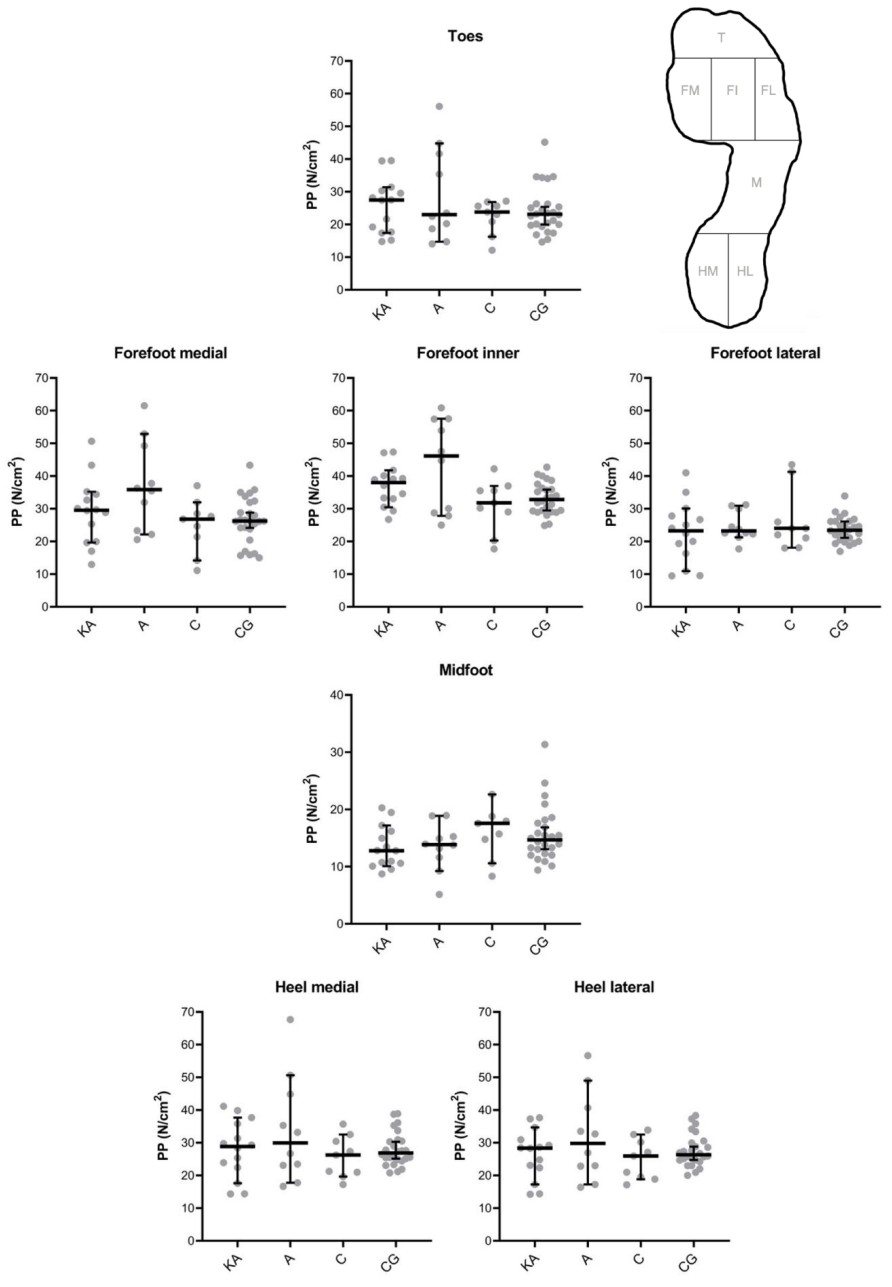

**Figure 3** **Peak Pressures (PPs) of the Keller's arthroplasty (KA), arthrodesis (A), cheilectomy (C) and control group (CG) in seven foot zones as shown in Fig. 1.** Results were presented as median and 95% confidence interval, with individual values. * All interventions were within each foot zone statistically compared to each other (*i.e.,* KA *vs.* A, KA *vs.* C, KA *vs.* CG, A *vs.* C, A *vs.* CG, C *vs.* CG) and no significant differences were detected ($P > 0.05$).

## DISCUSSION

The aim of this comparative study was to evaluate the long-term effects of Keller's arthroplasty, arthrodesis and cheilectomy on gait, plantar pressure distribution and clinical

**Table 3** Manchester-Oxford Foot Questionnaire (MOXFQ) results of the different surgical interventions[a][b].

|  | Keller's arthroplasty | Arthrodesis | Cheilectomy |
|---|---|---|---|
| Walking/standing | 24.1 ± 33.7 (0-89.3) | 21.4 ± 33.0 (0-71.4) | 21.4 ± 24.0 (0-57.1) |
| Pain | 21.3 ± 24.0 (0-70.0) | 16.5 ± 20.3 (0-50.0) | 22.2 ± 22.7 (0-65.0) |
| Social interaction | 31.3 ± 29.9 (0-100.0) | 23.8 ± 27.0 (0-68.8) | 35.4 ± 31.7 (0-100.0) |
| Index score | 25.0 ± 28.4 (0-81.3) | 20.5 ± 26.0 (0-62.5) | 25.2 ± 24.5 (0-70.3) |

Notes.
[a] Mean ± standard deviation presented and range in parentheses.
[b] Table represents the scores on each domain and the overall index score, whereas a score of 0 represents the best outcome and 100 the poorest outcome.
*All groups were statistically compared to each other and no statistically significant differences were detected ($P > 0.05$).

outcome of patients treated for hallux rigidus. The present study did not find differences based on effect-sizes and statistical analysis in spatio-temporal parameters nor for the plantar pressure analysis (PTIs and PPs) in foot zones and PROMs in patients treated for HR 22 years after surgery.

This is the first comparative pedobarographic study that evaluates Keller's arthroplasty, arthrodesis and cheilectomy as frequently performed surgical interventions for HR. Multiple studies have reported improvements in clinical outcome and patient satisfaction after all three surgical techniques (*Beertema et al., 2006*; *Coughlin & Shurnas, 2003b*; *Schneider et al., 2011*; *Stevens et al., 2017a*). For this reason, gait studies, such as pedobarographic studies are in particular relevant, since they can elaborate if satisfaction or complains after follow-up can be explained by locomotor alternations or inefficiencies (*Baker et al., 2016*; *Orlin & McPoil, 2000*). Pressure time integral and peak pressure are the most widely used variables for assessment of plantar loading in pedobarographic studies (*Keijsers, Stolwijk & Pataky, 2010*; *Melai et al., 2011*; *Orlin & McPoil, 2000*).

A limited number of studies are available evaluating plantar pressure of one of the three interventions (Keller's arthroplasty, arthrodesis, cheilectomy), while no study comparing all three interventions, is currently available (*Chraim et al., 2016*; *DeFrino et al., 2002*; *Gibson & Thomson, 2005*; *Lau & Daniels, 2001*; *Nawoczenski, Ketz & Baumhauer, 2008*; *Stevens et al., 2017b*). Four studies reported previously pedobarographic results after MTP1 arthrodesis (*Chraim et al., 2016*; *DeFrino et al., 2002*; *Gibson & Thomson, 2005*; *Stevens et al., 2017b*). Two of them compared pre- to postoperative differences and reported a statistically non-significant increase of PPs under the metatarsal heads (*DeFrino et al., 2002*; *Gibson & Thomson, 2005*). Two other studies evaluating plantar pressure postoperatively, reported a statistically significant increase in PP beneath the first metatarsal head compared to the contralateral foot after 47 months of follow-up (*Chraim et al., 2016*), while the other only detected a statistically significant increase in PP beneath the second to fifth metatarsal head compared to healthy controls after 27 months of follow-up (*Stevens et al., 2017b*). These studies show heterogeneity in methodological design, including the comparator groups and follow-up periods. Compared to the presented study are beneath the metatarsal heads (*i.e.,* medial and inner forefoot zone) higher pressures observed after MTP1 arthrodesis compared to healthy controls. However, the effects sizes of PTI are small and not significant different, indicating that total load exposure of a plantar area during

stance was comparable. Therefore, total load exposure during stance in a plantar foot zone seems to be similar to healthy controls.

Pedobarographic results after cheilectomy were reported by two studies previously and showed a slight but not significant increased pressures beneath the first and second metatarsal head and decreased pressure beneath the phalanx (*Lau & Daniels, 2001*; *Nawoczenski, Ketz & Baumhauer, 2008*). Besides, comparable results of both studies, differences in methodology remain, since one study compared the affected foot to the unaffected foot at 2 years follow-up (*Lau & Daniels, 2001*), while the other compared pre- to postoperative hallux rigidus results after 2 years follow-up (*Nawoczenski, Ketz & Baumhauer, 2008*). Therefore, study design (*i.e.,* longitudinal *vs.* cross-sectional) and comparator group remains different between both studies. Compared to the present study, also no differences in effect sizes nor significant differences in PTIs and PPs between cheilectomy subjects and healthy controls were observed.

Only one study evaluated pedobarograpic results after Keller's arthroplasty after 20-years of follow-up (*Schneider et al., 2011*). Solely plantar pressure data of the hallux and toes 2 to 5 were reported and results showed a decreased PP beneath the hallux and increased PP in toes 2 to 5, compared to healthy controls (*Schneider et al., 2011*). No statistical analyses were performed and plantar pressure data of the other zones were not reported. In the present study effect sizes were small and no statistically significant differences in PP and PTI beneath the toes were observed compared to healthy controls 22 years after surgery. Based on the present study results and results of previous published studies, it is suggested that Keller's arthroplasty, arthrodesis and cheilectomy as surgical interventions for HR does not have a major impact on plantar pressure distribution in the foot. After surgery, the biomechanics related to foot loading are not sufficiently influenced. While plantar pressure distribution is not sufficiently influenced, differences in compensation in kinematics and kinetics could still exist (*DeFrino et al., 2002*; *Nawoczenski, Ketz & Baumhauer, 2008*; *Stevens et al., 2017a*). Future studies are recommended to elucidated these effects and observe the impact on gait.

In the present study, no differences were observed in the foot-specific PROMs (MOXFQ) between the surgical interventions after 22 years of follow-up. Clinically important differences between the groups were below the minimally clinically important difference (MCID) values of: 16 for walking/standing, 12 for pain and 24 for social interaction domain (*Dawson et al., 2007*). Four studies have previously investigated the MOXFQ after HR surgery. Compared to the present study, Keller's arthroplasty showed better outcomes after 6-months follow-up (*Maher, 2017*). Also in cheilectomy after 17 months follow-up (*Harrison et al., 2010*) were better outcomes observed, while comparable results were detected after 50 months follow-up (*Teoh et al., 2019*). However, better results were observed in the present study for arthrodesis compared to a previous arthrodesis study with 10-months follow-up (*Fanous, Ridgers & Sott, 2014*). The results suggest that cheilectomy and Keller's arthroplasty showed better outcomes at short term, while improved outcomes after arthrodesis are observed at long-term follow-up. A comparable trend was observed within the original study performed after 7 years of follow-up (*Beertema et al., 2006*). The results of the present study indicate that patients function is similar 22 years after surgery,

irrespective to the performed surgery. This supports the finding that also no difference in spatio-temporal gait parameters and pedobarographic assessment were observed. This is an interesting observation, as for the last few decades there has been an increasing preference to perform cheilectomies and arthrodesis over Keller's arthroplasty in HR patients. The preferences raised due to the fact that Keller's arthroplasty results in a nonfunctional first ray, which is basically the cause of several complications such as, cock-up deformity, limitation of active flexion and a floppy toe (*Coutts, Kilmartin & Ellis, 2012*). Therefore, Keller's arthroplasty was less favorable for the fear of having a nonfunctional first ray, which could result in pain and functional limitations (*DeFrino et al., 2002*; *Gibson & Thomson, 2005*).

Correlation analysis was performed between pedobarographic results and clinical outcome. With the numbers available, no substantial associations were observed comparing PTI forefoot and the MOXFQ domains in each intervention group. Therefore, no relation between PROMs and pedobarographic results was observed. Predicting plantar pressure in the forefoot based on PROMs and vice versa is currently not reliable for clinical practice based on the present study results.

Besides the assessment after long-term follow-up evaluation, pedobarographically and clinically of the three most commonly used interventions for HR, we acknowledge that this study has some limitations. A limited number of subjects per intervention were available for follow-up resulting in a small sample size. All available participants at follow-up were included, however a substantial part of the initial cohort was not available for follow-up. As already mentioned, 23% of patients was deceased, 8% was loss to follow-up and 32% was not able or willing to participate. The high drop-out rate is a commonly reported problem in HR follow-up studies, since HR mainly affects middle-aged to elderly individuals, therefore a substantial part of the patients may be deceased at long-term follow-up (*Coughlin & Shurnas, 2003a*; *Ho & Baumhauer, 2017*; *Stevens et al., 2017a*). Furthermore, subjects should be able to walk individually barefooted in gait studies, which is an additional demand for elderly individuals participating in this kind of studies. This was also the case in the present study, since a major part (32% of patients), was not willing or able to participated since they did not walk or could not walk individually due to their progressed age. Based on the limited number of subjects in the present long-term follow-up study, underpowering cannot be ruled out. Therefore, effect sizes were reported to present the magnitude of observed differences between the groups and to facilitated the decision whether a clinically relevant effect was found (*Aarts, Van den Akker & Winkens, 2014*; *Sullivan & Feinn, 2012*). To accompany the effect sizes, statistical analyses was additionally performed. Furthermore, no major foot deviations or additional foot surgery was performed until follow-up evaluation. This study was limited in the evaluation of other comorbidities at the musculoskeletal system, which are not uncommon in this elderly population evaluated after long-term follow-up. Despite the limitations, this study is of additive value since it evaluates gait, plantar pressure and PROMs of the three most performed surgical interventions for HR after a very long-term after surgery.

## CONCLUSION

The present study, showed trends of comparable effect sizes, without clinically relevant differences on spatio-temporal parameters, plantar pressure analysis and PROMs after Keller's arthroplasty, arthrodesis and cheilectomy 22 years after surgery. Proper preoperative staging of HR is essential since cheilectomy is predominantly recommended for patients with mild to moderate HR (grade 1 and 2) and Keller's arthroplasty and arthrodesis is predominately used in more progressed HR (grade 2 and 3). The present study results suggest that Keller's arthroplasty, arthrodesis and cheilectomy are appropriate surgical treatments for a selective group of patients suffering from a symptomatic hallux rigidus and patients function to a comparable level after long-term follow-up. Further research is recommended to approve the observation in the present study after such long-term follow-up.

### Funding
The authors received no funding for this work.

### Competing Interests
Thijs Smeets is the owner of Smeets Loopcomfort and was involved as researcher. Thijs Smeets and his company do not have financial or other competing interests with the performed research. The other authors declare that they have no competing interests.

### Author Contributions
- Robin de Bot conceived and designed the experiments, performed the experiments, analyzed the data, prepared figures and/or tables, authored or reviewed drafts of the article, and approved the final draft.
- Jasper Stevens conceived and designed the experiments, performed the experiments, analyzed the data, prepared figures and/or tables, authored or reviewed drafts of the article, and approved the final draft.
- Thijs Smeets conceived and designed the experiments, performed the experiments, analyzed the data, authored or reviewed drafts of the article, and approved the final draft.
- Adhiambo Witlox conceived and designed the experiments, analyzed the data, authored or reviewed drafts of the article, and approved the final draft.
- Wieske Beertema conceived and designed the experiments, analyzed the data, authored or reviewed drafts of the article, and approved the final draft.
- Roel Hendrickx conceived and designed the experiments, analyzed the data, authored or reviewed drafts of the article, and approved the final draft.
- Kenneth Meijer conceived and designed the experiments, analyzed the data, authored or reviewed drafts of the article, and approved the final draft.
- Martijn Schotanus conceived and designed the experiments, performed the experiments, analyzed the data, authored or reviewed drafts of the article, and approved the final draft.

## Human Ethics

The following information was supplied relating to ethical approvals (*i.e.*, approving body and any reference numbers):

Approval for this study was obtained from the local ethics committee Zuyderland Medical Centre (METC Z, Nr. 17T09), and all patients provided written informed consent.

## Data Availability

The information of the study population is available in the Supplemental File.

## Supplemental Information

Supplemental information for this article can be found online at http://dx.doi.org/10.7717/peerj.16296#supplemental-information.

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
