# Peer review of "Gait and dynamic pedobarographic analyses in hallux rigidus patients treated with Keller’s arthroplasty, arthrodesis or cheilectomy 22 years after surgery"

_PeerJ, doi:10.7717/peerj.16296_

## Round 0.1 · original submission · Major Revisions

The main considerations are related to the state of the art that supports the research problem and the experimental design (eg, influence of only using flat surface, follow-up concept). The latter may impact the generalizability of the results. For these reasons, the report needs extensive review and reassessment by the reviewers. For more details, please review the reviewers' comments.

Reviewer 1 ·

Basic reporting

The aim of this study was to evaluate the effects of Kellerís arthroplasty, arthrodesis and cheilectomy on gait, plantar pressure distribution and clinical outcome of patients treated for HR after 22-years of follow-up.
This study is well written and give an insight into the long-term assessment of Kellerís arthroplasty, arthrodesis and cheilectomy on gait patterns.
My major consent about the study is the concept of “follow up”. The authors only did one measure over time. This doesn't allow tracking changes over time. A more appropriate term might be considered, for instance, "long-term assessment". In addition, the small sample size may limit the conclusion. The authors need to consider if the results may not be a statistical error type II. Also, the authors need to consider adjusting the results with covariates such as walking velocity, BMI, and other potential confounders variables.

Experimental design

The authors conducted the measurements with participants walking barefoot during the trials, which may not replicate real-world conditions. Additionally, the study involved walking on a flat surface only. A single inclination could limit the generalizability of the results. Including uphill and downhill surfaces or run, conditions may give extra information about the locomotion patterns and differences. One limitation is that the protocol might only partially apply to real-life scenarios. Other simple walking tests, such as the 6-minute walk test, could also be beneficial to implement.

Validity of the findings

Specific comments.
Regarding the correlation analysis, these results need to be linked with the aim of the study.
Line 125. The authors need to describe better the familiarization time regarding the protocol used.
In the results section, the effect size as the Hodges-Lehmann estimator is a type of effect size; its interpretation differs from traditional effect sizes like Cohen's. The Cohen effect size may help to interpret the results better.

Additional comments

References to consider:
Ho IJ, Hou YY, Yang CH, Wu WL, Chen SK, Guo LY. Comparison of Plantar Pressure Distribution between Different Speed and Incline During Treadmill Jogging. J Sports Sci Med. 2010 Mar 1;9(1):154-60.
Damavandi, M., Dixon, P. C., & Pearsall, D. J. (2010). Kinematic adaptations of the hindfoot, forefoot, and hallux during cross-slope walking. Gait & posture, 32(3), 411-415.
Damavandi, M., Dixon, P. C., & Pearsall, D. J. (2012). Ground reaction force adaptations during cross-slope walking and running. Human movement science, 31(1), 182-189.

·

Basic reporting

no comment

Experimental design

The authors should go deeper into what is the knowledge gap on the subject, beyond just decreeing that there is no information on long-term effects.

Validity of the findings

There is a need for transparency regarding the processing codes used in part of the data analysis.

---

## Round 0.2 · accepted · Accept

The authors have satisfactorily addressed all the comments of the reviewers. For these reasons, the manuscript is accepted to be published. Congratulations.

Reviewer 1 ·

Basic reporting

The authors have addressed my inquiries and made the necessary modifications accordingly.

Experimental design

No comments

Validity of the findings

No comments

Additional comments

No comments

·

Basic reporting

no comment

Experimental design

no comment

Validity of the findings

no comment